# DJ-1 Expression Might Serve as a Biologic Marker in Patients with Bladder Cancer

**DOI:** 10.3390/cancers14102535

**Published:** 2022-05-21

**Authors:** Shuhei Hirano, Kazumasa Matsumoto, Kei Tanaka, Noriyuki Amano, Dai Koguchi, Masaomi Ikeda, Yuriko Shimizu, Benio Tsuchiya, Ryo Nagashio, Yuichi Sato, Masatsugu Iwamura

**Affiliations:** 1Department of Urology, School of Medicine, Graduate School of Medical Sciences, Kitasato University, Sagamihara 252-0374, Japan; s.hirano@med.kitasato-u.ac.jp (S.H.); dm19002@st.kitasato-u.ac.jp (N.A.); dai.k@med.kitasato-u.ac.jp (D.K.); ikeda.masaomi@grape.plala.or.jp (M.I.); yulico@med.kitasato-u.ac.jp (Y.S.); yuichi@med.kitasato-u.ac.jp (Y.S.); miwamura@med.kitasato-u.ac.jp (M.I.); 2Department of Applied Tumor Pathology, Graduate School of Medical Sciences, Kitasato University, Sagamihara 252-0374, Japan; keitanaka@nagasaki-u.ac.jp (K.T.); benio@med.kitasato-u.ac.jp (B.T.); nagashio@kitasato-u.ac.jp (R.N.); 3Department of Pathology, Graduate School of Biomedical Sciences, Nagasaki University, Nagasaki 852-8501, Japan

**Keywords:** bladder cancer, urothelial carcinoma, DJ-1, serum, cystectomy

## Abstract

**Simple Summary:**

Although serum DJ-1 has been evaluated in some types of cancer, it has not been analyzed in detail in bladder cancer (BC). Furthermore, DJ-1 expression patterns in BC and their clinicopathologic significance and relationship with prognosis are unclear. In the present study, we evaluated serum DJ-1 levels in BC and the localization of DJ-1 expression in BC tissues. Our research showed that serum DJ-1 was significantly higher in 172 patients with BC who underwent the transurethral resection of bladder tumors (TURBT) than in those with urolithiasis or in healthy participants. Immunohistochemically, a DJ-1 cytoplasm-positive (Cy+) and nucleus-negative (N−) pattern in 92 archived radical cystectomy BC specimens was associated with a significantly increased risk for lower overall, recurrence-free and cancer-specific survival. Our findings suggest that DJ-1 might be a new biomarker for diagnosing BC and predicting biologically aggressive cancers so as to determine the appropriate treatment modality after radical cystectomy.

**Abstract:**

The overexpression of DJ-1 protein and its secretion into the bloodstream has been reported in various neoplasms. However, serum levels and the subcellular localization of DJ-1 have not been analyzed in detail in bladder cancer (BC). Our comprehensive analysis of these variables started with the measurement of DJ-1 in serum from 172 patients with BC, 20 patients with urolithiasis and 100 healthy participants. Next, an immunohistochemical study of DJ-1 expression and localization was conducted in 92 patients with BC, and associations with clinicopathologic factors and patient outcomes were evaluated. Serum DJ-1 was significantly higher in patients with BC than in those with urolithiasis or in healthy participants. Immunohistochemically, a cytoplasm-positive (Cy+) and nucleus-negative (N−) DJ-1 pattern was associated with age and pathologic stage. Log-rank tests indicated that the Cy+, N− pattern was significantly associated with overall survival (OS), recurrence-free survival (RFS), and cancer specific survival (CSS). In addition, the Cy+, N− pattern was an independent prognostic factor in the multivariate analysis adjusted for the effects of the clinicopathologic outcomes. The investigation of DJ-1 expression might help physicians to make decisions regarding further follow-up and additional treatments.

## 1. Introduction

Bladder cancer (BC) is the 10th most common cancer in the world [1], and 80% of patients present with non-muscle-invasive bladder cancer (NMIBC). In up to 50% of NMIBC cases, the disease eventually recurs despite transurethral resection with the intravesical instillation of bacillus Calmette–Guérin, and in up to 30%, the disease progresses to muscle-invasive BC [2]. The “gold standard” for the initial diagnosis of BC remains cystoscopy coupled with urine cytology [3]; however, cystoscopy is unpleasant for patients and costly for healthcare systems [4]. Although urine cytology is the most common technique for the diagnosis and surveillance of BC, it provides only 27% sensitivity in low-grade carcinoma [5]. Therefore, to reduce the need for cystoscopy, many studies have attempted to improve the diagnostic performance of biomarkers and find new diagnostic modalities.

The US Food and Drug Administration has approved six urinary assays for clinical use in conjunction with cystoscopy. The NMP22 enzyme-linked immunosorbent assay (ELISA), the NMP22 BladderChek (Abbott Laboratories, Chicago, IL, USA) and the UroVysion kit (Abbott Laboratories) have Food and Drug Administration approval for diagnosis and surveillance, while the ImmunoCyt/uCyt+ (DiagnoCure, Quebec City, QC, Canada), BTA-TRAK (Bard Diagnostic Sciences, Murray Hill, NJ, USA) and BTA-STAT (Polymedco, Cortlandt Manor, NY, USA) tests are all approved for bladder surveillance after the diagnosis of a primary tumor. The FDA-approved biomarkers have high false positive rates, and their low specificity remains one of the greatest limitations of urine biomarkers in clinical practice [6].

Several urinary biomarkers have been developed for the follow-up of patients with NMIBC [7,8]. However, most of the available urinary biomarkers are characterized by low positive predictive values that limit their application in routine clinical practice [7]. Although efforts have been made to identify serum-based biomarkers for BC detection, such technologies are unfortunately not well developed. Furthermore, no blood-based tests for detection are currently available [4,9,10]. With respect to proteomics analyses, we have previously reported several BC-related proteins, and here we focus on DJ-1 [11,12,13].

DJ-1 was first reported as an oncogene that can transform N1H3T3 cells in cooperation with activated *RAS* [14]. A conserved protein, DJ-1 is coded by *PARK7* and is ubiquitously present in cells, where it is involved in diverse cell processes such as cell transformation, the control of protein–RNA interactions, and the prevention of cell death from oxidative stress-induced apoptosis [15]. High levels of DJ-1 expression have been reported during initiation and progression in some types of cancer [16,17]. Serum DJ-1 is already used as a diagnostic marker for lung, breast and pancreatic cancer [16,17,18].

Several studies have evaluated DJ-1 as a urinary biomarker for the detection of BC [19,20], but urinary protein biomarkers often vary with grade and stage—that is, they have lower sensitivity in low-grade and low-stage cancers. Furthermore, we firmly believe that urine analysis is affected by renal function and urine volume. To our knowledge, no studies have demonstrated that DJ-1 acts as a serum marker for the detection of bladder cancer.

In the present study, we aimed to determine whether DJ-1 is a useful serum biomarker for the diagnosis of urothelial carcinoma (UC) and whether the immunohistochemical expression of DJ-1 protein in patients treated with radical cystectomy can be used to assess the progression of BC.

## 2. Materials and Methods

### 2.1. Patients

We retrospectively reviewed preoperative serum samples from 293 patients who underwent the transurethral resection of bladder tumors at the Kitasato University Hospital between 2009 and 2015. The 172 patients included in the study excluded those who had cancers other than UC, duplicate cases, and those who had histologic variants or carcinoma in situ (Table 1). This study also enrolled 20 patients with urolithiasis and 100 healthy participants as controls. Patients with urolithiasis had been diagnosed with any or all of the following: bladder stones, kidney stones or ureteral stones. Healthy participants had no past or current diseases, including cancer, urolithiasis, autoimmune, inflammatory or metabolic diseases. Sera from all study participants were kept in our laboratory at −80 °C until use.

We checked DJ-1 immunoreactivity in another cohort of 92 consecutive patients with BC (72 men, 20 women; age: 40–81 years (median: 65 years; mean: 63.5 years)) who underwent radical cystectomy with pelvic and iliac lymphadenectomy at Kitasato University Hospital between 1990 and 2011. All patients underwent an open or laparoscopic procedure with regional lymphadenectomy [21]. No patient received chemotherapy or radiotherapy before surgery.

The indications for cystectomy in patients with initial NMIBC included intravesical therapy failure or muscle-invasive BC. The 2002 TNM classification was used for pathology staging and the 1973 World Health Organization classification was used for pathology grading. Patients were considered positive for lymphovascular invasion (LVI) if cancer cells were present in the endothelial space. When cancer cells had merely invaded the vascular lumen, patients were considered LVI-negative. Formalin-fixed, paraffin-embedded blocks representing the most invasive areas of each tumor were collected for further investigation.

Each patient was scheduled for postoperative follow-up every 3 months for the first year, twice during the second year, and once per year thereafter. More frequent examinations were scheduled if clinically indicated. After surgery, 16 patients (17.4%) received adjuvant chemotherapy (methotrexate, vinblastine, doxorubicin, cisplatin, gemcitabine and cisplatin) because of adverse pathologic characteristics, including regional or distant lymph node metastases or involvement. Upon disease recurrence, 13 patients (14.1%) received one of two chemotherapeutic regimens. The study protocol was conducted according to the guidelines of the Declaration of Helsinki and was approved by the Institutional Review Board of Kitasato University School of Medicine and Hospital (nos. B17-010 and B18-149). Informed consent for the serum analysis was obtained from all study participants. Information about the opportunity to opt out of the immunohistochemical analysis was provided to participants on our website and using posters. Patients could refuse study entry and discontinue participation at any time.

### 2.2. Measurement of Serum DJ-1

Serum DJ-1 was detected using the reverse-phase protein array analysis described by Kobayashi et al., with some modifications [22]. The reverse-phase protein array requires significantly smaller amounts of clinical samples for quantification than established clinical tests, such as ELISA [23]. We have been conducting and reporting on reverse-phase protein array analysis using micro-dot blot [22,24]. Serum samples were diluted 1:15 with 0.01% TritonX-100/phosphate-buffered saline without divalent ions (PBS−/T) and spotted in quadruplicate onto high-density amino-group-induced glass slides treated with dimethyl sulfoxide (SDM0011 (Matsunami Glass, Osaka, Japan)). A glass slide microarrayer (VP 478A (V&P Scientific, San Diego, CA, USA)) was used to achieve a 640-spot-per-slide format. Recombinant DJ-1 tagged with glutathione S-transferase was prepared in a wheat germ cell-free system [25].

Various concentrations (0.156, 0.312, 0.625, 1.25, 2.5, 5.0 and 10.0 ng/µL) of DJ-1 were also used to obtain a calibration curve. All spotted slides were immobilized in a moist chamber for 48 h at room temperature. After blocking with 0.5% casein solution (0.5% casein in PBS−/T) for 1 h at room temperature, the slides were incubated with 100-times-diluted anti-DJ-1 monoclonal antibody (Clone 3E8 (MBL, Nagoya, Japan)) for 16–18 h at 4 °C. After rinsing three times in PBS−/T for 5 min each time, slides were reacted with 100-times-diluted biotinylated anti-mouse immunoglobulin G (BA-2000 (Vector Laboratories, Burlingame, CA, USA)) for 1 h at 37 °C. The sections were then rinsed three times in PBS−/T for 5 min each time and subsequently incubated with 1000-times-diluted conjugated streptavidin–horseradish peroxidase (GE Healthcare Bio-Sciences, Pittsburgh, PA, USA) for 30 min at 37 °C. After rinsing three times in PBS−/T for 5 min each time, the slides were allowed to react with 200-times-diluted Cy5-conjugated tyramide (PerkinElmer Life Sciences, Massachusetts, MA, USA) for 20 min at 37 °C.

After rinsing three times in PBS−/T for 5 min each time, the slides were rinsed twice with ultrapure deionized water. The slides were read using a GenePix 4000 B microarray scanner (Molecular Devices, San Jose, CA, USA). The relative fluorescence intensity of each sample spot was quantified using GenePix Pro 6.0 software application (Molecular Devices).

### 2.3. Immunohistochemistry and Scoring

Three 3-micrometre-thick sections of formalin-fixed, paraffin-embedded tissue were deparaffinized in xylene and rehydrated using a descending ethanol series and tap water. After treatment with 3% hydrogen peroxide for 10 min, the sections were antigen-retrieved by autoclaving in 1 mM EDTA·2Na and 0.01 mol/L Tris–HCl buffer (pH 9.0) at 121 °C for 10 min. After cooling to room temperature and blocking with 0.5% casein for 10 min, the sections were allowed to react with anti-DJ-1 monoclonal antibody (Clone 3E8 (MBL)) at 37 °C for 2 h. After rinsing three times in Tris-buffered saline for 5 min each time, the sections were allowed to react with a ChemMate EnVision detection kit (Dako, Glostrup, Denmark) at room temperature for 30 min. The sections were subsequently visualized using a stable DAB solution (Life Technologies, Carlsbad, CA, USA) and counterstained with Mayer hematoxylin. All immunostained sections were reviewed by two investigators (S.H. and Y.S.) who were blinded to the clinical and pathology data. Disagreements about the cases were reviewed and discussed until a consensus was reached.

Nuclear and cytoplasmic DJ-1 staining of tumor cells was considered positive. In non-neoplastic tissues, nuclear staining was observed in urothelial cells and peritumoral vascular endothelial cells, which were used as internal positive controls. Using those internal control cells, staining intensity in the nucleus was categorized into either a negative group (no positive cells or weaker staining than in the positive control cells) or a positive group (staining the same as or stronger than the positive control cells) [26]. The staining intensity of the cytoplasm was scored on a scale of 0 to 3: 0—no staining; 1—weak staining; 2—moderate staining; 3—strong staining. The percentage of tumor cells expressing DJ-1 was calculated and multiplied by the staining intensity score over an average of three areas to obtain a semi-quantitative H score (maximum value: 300, corresponding to 100% of tumor cells being DJ-1-positive, with an overall staining score of 3). DJ-1 cytoplasmic expression scores were categorized as positive (≥170) or negative (<170), based on an average score of 170.

### 2.4. Statistical Analyses

Serum DJ-1 levels in healthy participants and patients with BC or urolithiasis were compared using the Mann–Whitney *U*-test. A receiver operating characteristic curve was generated by plotting the sensitivity against the false-positive rate (100—specificity). The optimal cutoff value based on the sensitivity and specificity of serum DJ-1 was estimated using the Youden index [27].

In the immunohistochemistry analysis, age (<65 vs. ≥65 years), pathology stage (≤T1 vs. ≥T2), grade (1 and 2 vs. 3) and lymph node status (N0 vs. N1 and N2) were evaluated as dichotomized variables. The chi-squared test was used to evaluate associations between sex, age, pathology stage, pathology grade, carcinoma in situ, lymph node status and LVI. The Kaplan–Meier method was used to calculate survival, and differences were assessed using the log-rank statistic. Multivariate survival analyses were performed using a Cox proportional hazards regression model controlling for DJ-1 expression, pathology stage, pathology grade, LVI and lymph node metastases.

Statistical significance was set at *p* < 0.05. All reported *p* values are two-sided. All analyses were performed using the Stata software application (version 15 for Mac: StataCorp LLC, College Station, TX, USA).

## 3. Results

### 3.1. Serum DJ-1

Figure 1 shows the serum DJ-1 concentrations in each study group. In the BC group, the median DJ-1 concentration was 21.93 ng/mL (range: 0.00–235.06 ng/mL), which was significantly higher than the concentrations in the urolithiasis and healthy participant groups (median: 9.67 ng/mL [range: 4.04–20.37 ng/mL] and 6.58 ng/mL [0.75–30.39 ng/mL], respectively, each *p* < 0.001).

Serum DJ-1 in patients with pTa/1 BC (median: 23.5 ng/mL; range: 2.3–235.1 ng/mL) was significantly higher than it was in patients with pT2 BC (median: 15.2 ng/mL; range: 0.00–207.00 ng/mL; *p* = 0.009). Furthermore, the results of comparing serum DJ-1 in patients with pTa/1 BC and pT2 BC directly with the serum of healthy participants and patients with urolithiasis were consistent (Appendix A).

In the receiver operating characteristic analysis, serum DJ-1 levels were significantly higher in patients with BC than in those with urolithiasis (area under the curve (AUC), 0.83; *p* < 0.001, Figure 2A) or in healthy participants (AUC, 0.88; *p* < 0.001; Figure 2B). Moreover, serum DJ-1 levels were significantly higher in patients with NMIBC than in healthy participants (AUC, 0.92; *p* < 0.001; Figure 2C).

### 3.2. DJ-1 Expression by Immunohistochemistry and Clinicopathologic Outcomes

For this study, we grouped DJ-1 staining into four patterns based on localization: cytoplasm positive and nucleus negative (Group 1, *n* = 28, Figure 3B); cytoplasm positive and nucleus positive (Group 2, *n* = 26, Figure 3C); cytoplasm negative and nucleus positive (Group 3, *n* = 12, Figure 3D); and cytoplasm negative and nucleus negative (Group 4, *n* = 26, Figure 3E). In a pilot analysis, overall survival (OS) was significantly shorter for patients in group 1 than for participants in the other groups (*p* < 0.001, Figure 4).

In the present study, the expression level of DJ-1 in tumor tissues alone (independently of its localization) was not associated with significant differences in OS and cancer-specific survival (CSS (*p* = 0.51 and 0.53, respectively, Appendix A)). The expression of DJ-1 in the cytoplasm was not associated with either OS or CSS (*p* = 0.15 and 0.19, respectively, Appendix A). In contrast, the expression of DJ-1 in the nucleus was associated with OS (*p* = 0.029, Appendix A), but it was not associated with CSS (*p* = 0.11, Appendix A). Based on these observations, we compared Group 1 with Groups 2–4.

Table 2 shows the correlation between DJ-1 expression and the clinicopathologic characteristics of the participants. In Group 1, DJ-1 expression was associated with age (*p* = 0.011), pathology stage (*p* = 0.025) and lymph node status (*p* = 0.039). Sex, pathology grade, carcinoma in situ status and LVI status did not differ significantly between Group 1 and Groups 2–4.

At a median follow-up of 33.7 months, 47 patients (51.1%) experienced disease recurrence, 42 (45.7%) died of BC, and 11 (12.0%) died of other causes. Figure 5 shows the association between DJ-1 expression and patient survival. Compared with the other groups, group 1 showed significant differences in OS, recurrence-free survival and CSS (*p* < 0.001, *p* = 0.03 and *p* = 0.009, respectively). The cumulative OS rates for patients in Group 1 and Groups 2–4 were 64.1% and 83.9% at one year, 34.4% and 75.6% at two years, and 25.8% and 60.7% at five years, respectively.

In Cox’s proportional hazards regression analysis, DJ-1 expression and lymph node metastasis were associated with OS, recurrence-free survival and CSS (Table 3).

## 4. Discussion

This study has demonstrated that serum DJ-1 in patients with BC is elevated and that its subcellular localization is associated with poor prognosis. DJ-1 is a 19.8 kDa protein that is present in both the nucleus and the cytoplasm. It is conceivable that DJ-1 can activate protein kinase B (PKB/Akt) signaling by suppressing PTEN, resulting in the progression of various carcinomas [17,28]. However, the mechanism of the DJ-1 protein remains to be clarified.

The noninvasive detection and monitoring of BC remains challenging. Novel panels of urinary proteins have been developed as biomarkers for the diagnosis of primary BC. Rosser et al. reported that seven biomarkers (interleukin 8, matrix metalloproteinase 9, PA1, angiogenin, vascular endothelial growth factor, matrix metalloproteinase 10 and apolipoprotein E) had an AUC of 0.88, 74% sensitivity, and 90% specificity [29]. Kumar et al. showed that ELISA, using five urinary biomarkers (coronin-1A, apolipoprotein A4, semenogelin-2, gamma-synuclein and DJ-1) had an AUC of 0.92 (sensitivity 79.2%, specificity 100%) in diagnosing Ta/1 BC [30]. Soukup et al. showed that diagnostic accuracy was low for urinary DJ-1 alone (sensitivity—61.1%; specificity—87.7%; AUC—0.64) [19]. It is conceivable that urinary DJ-1 on its own is insufficient as a diagnostic marker for BC.

Notably, elevated serum DJ-1 has also been found in patients with lung, breast and pancreatic cancer. Han et al. showed that serum DJ-1 could differentiate early-stage lung cancer from benign diseases [16]. Kawate et al. reported that a cancer-specific DJ-1 isoform in serum clearly distinguishes patients with breast cancer from individuals with non-cancerous lesions, even when serum DJ-1 is highly elevated [17]. He et al. reported that the AUC, sensitivity and specificity for serum DJ-1 were higher than those for carbohydrate antigen 19-9 in patients with pancreatic cancer [18]. However, little is known about serum markers, including DJ-1, in patients with BC. To the best of our knowledge, the present study is the first to examine serum DJ-1 levels in patients with BC.

We found that serum levels of DJ-1 were higher in patients with BC than in those with urolithiasis or in healthy participants. In addition, we observed that serum levels of DJ-1 were higher in patients with NMIBC than in those with muscle-invasive BC. Han et al. reported that serum concentrations of DJ-1 were lower in lung cancer with distant metastases than in lung cancer with focal lesions, suggesting that the downregulation of DJ-1 (*PARK7*) occurs during metastasis [16]. They suggested that DJ-1 could potentially be a biomarker for early diagnosis and the monitoring of lung cancer metastasis. Although the precise mechanism remains unclear, we speculate that DJ-1 downregulation might occur during tumor progression.

In terms of immunohistochemistry, DJ-1 has been reported to be overexpressed in many types of cancer, and other authors have also proposed it as a prognostic biomarker [16,17,18]. DJ-1 overexpression has been reported in invasive UC, but the subcellular localization of DJ-1 staining in UC is poorly documented [31]. Miyajima et al. showed that nuclear DJ-1 staining was reduced in malignant astrocytomas compared with benign tumors and normal astrocytes; however, the accumulation of cytoplasmic DJ-1 protein was also observed [26]. These findings accord with our observations in the present study. Although DJ-1 subcellular localization varied in UC tissues, the overexpression in the cytoplasm and the low expression in the nucleus in surgical specimens were associated with pathology stage and shortened survival.

The overexpression of DJ-1 is associated with angiogenesis, the regulation of glycolytic metabolism, and the inhibition of apoptosis [32]. In invasive breast cancer, the increased expression of cytoplasmic DJ-1 has been observed, while *PTEN* on chromosome 10 is absent [28]. *PTEN* is one of the most frequently mutated tumor suppressor genes in various types of cancer. Lee et al. report that the overexpression of DJ-1 and the loss of *PTEN* are associated with invasive UC [31]. Based on these results, the overexpression of cytoplasmic DJ-1 might be correlated with the deletion of *PTEN* or a reduction in its expression. Overexpression in the cytoplasm and low expression in the nucleus in BC tissues might be a result of apoptosis attributable to *PTEN* downregulation by long non-coding RNAs such as LINC00641 or DUXAP8 [33]. The roles of DJ-1 and *PTEN* constitute a possible explanation for the association between increased cytoplasmic staining and decreased nuclear staining and poor prognosis in BC. These observations suggest that DJ-1 expression could serve as a factor in the evaluation of adjuvant chemotherapy or immune checkpoint inhibitor treatment.

Although DJ-1 expression was associated with clinicopathologic findings regardless of whether serum or tissue samples were evaluated, the present study has several limitations. First, we could not estimate prognosis based on DJ-1 as a serum biomarker. Second, any evaluation of serum DJ-1 must take note of the fact that its concentration is likely to be increased in patients with other cancers [16,18]. Third, the retrospective design and small sample size of our study might have limited the statistical analyses. Large prospective studies are needed to clarify the value of DJ-1 for detection and outcome prognostication in BC. Fourth, serum DJ-1 was measured using only reverse-phase protein array analysis. It will be necessary to compare that approach with other measurement methods such as Western blotting and ELISA, considering the need for the stable measurement of DJ-1 in clinical application. Fifth, different patient cohorts were used for the serum and immunohistochemistry analyses. A better understanding of any associations of serum DJ-1 with related immunohistochemistry findings remains to be established. Finally, BC management by radical cystectomy and choice of postoperative treatment (such as adjuvant chemotherapy) was decided by a number of individual surgeons, a factor that might have influenced patient outcomes.

## 5. Conclusions

Serum DJ-1 was higher in patients with BC than in patients with urolithiasis or in healthy participants. In addition, the localization of DJ-1 in BC tissues was associated with established features of biologically aggressive BC, such as advanced stage with lymph node involvement. DJ-1 expression might be useful for helping physicians decide on further follow-up and additional treatments for their patients. Further genomic and proteomic studies are required to clarify the role of DJ-1 in BC.

## Figures and Tables

**Figure 1 cancers-14-02535-f001:**
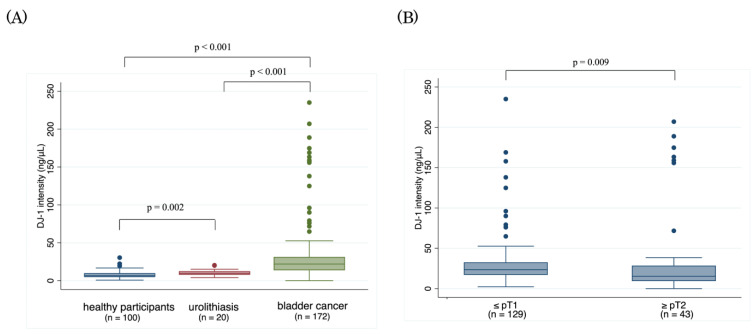
Serum concentrations of DJ-1 protein by reverse-phase protein array analysis (**A**) in patients with bladder cancer or urolithiasis and in healthy participants, and (**B**) in patients with non-muscle-invasive bladder cancer and muscle-invasive bladder cancer.

**Figure 2 cancers-14-02535-f002:**
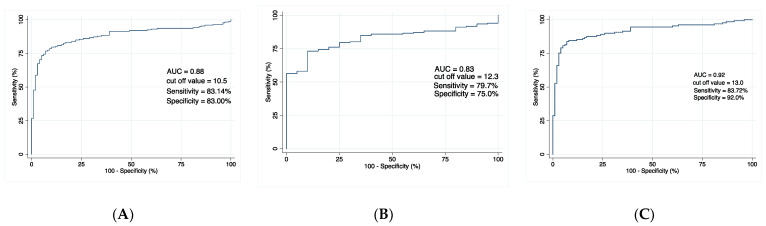
Receiver operating characteristic analysis of preoperative serum concentrations of DJ-1 for the detection of bladder cancer in (**A**) patients with bladder cancer compared with healthy participants, (**B**) patients with bladder cancer compared with patients with urolithiasis, and (**C**) patients with Ta/1 bladder cancer compared with healthy participants. AUC = area under the curve.

**Figure 3 cancers-14-02535-f003:**
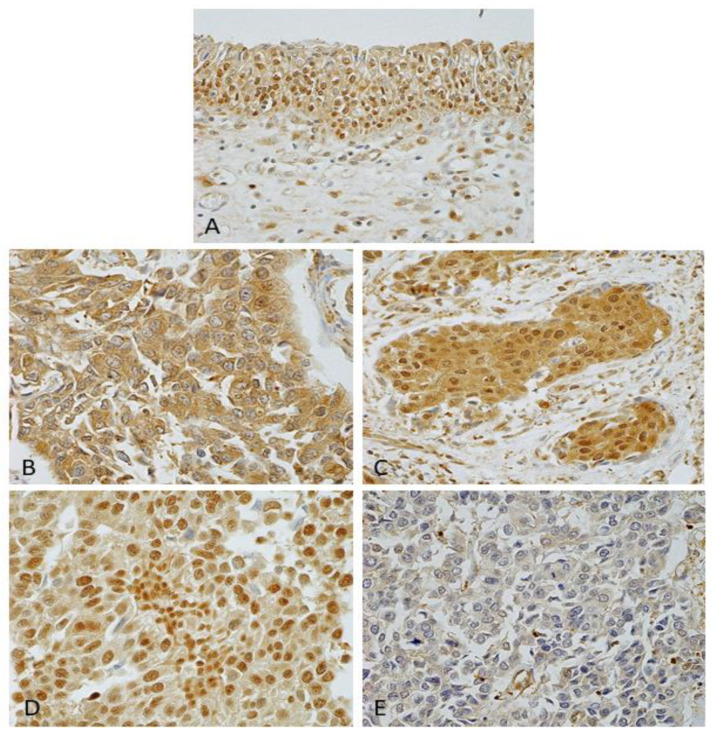
DJ-1 expression patterns in non-neoplastic urothelial cells and urothelial carcinoma. (**A**) Nuclear staining observed in non-neoplastic urothelial cells. (**B**) Cytoplasm-positive, nucleus-negative: Group 1. (**C**) Cytoplasm-positive, nucleus-positive: Group 2. (**D**) Cytoplasm-negative, nucleus-positive: Group 3. (**E**) Cytoplasm-negative, nucleus-negative: Group 4. All 400× original magnification.

**Figure 4 cancers-14-02535-f004:**
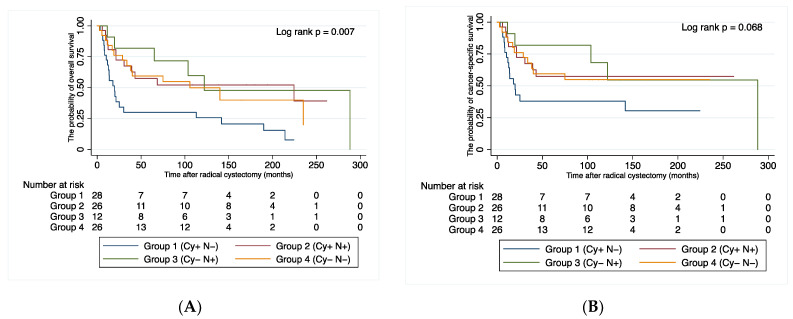
Probability of (**A**) overall survival and (**B**) cancer-specific survival after radical cystectomy by DJ-1 expression group. Group 1—cytoplasm-positive, nucleus-negative; Group 2—cytoplasm-positive, nucleus-positive; Group 3—cytoplasm-negative, nucleus-positive; Group 4—cytoplasm-negative, nucleus-negative.

**Figure 5 cancers-14-02535-f005:**
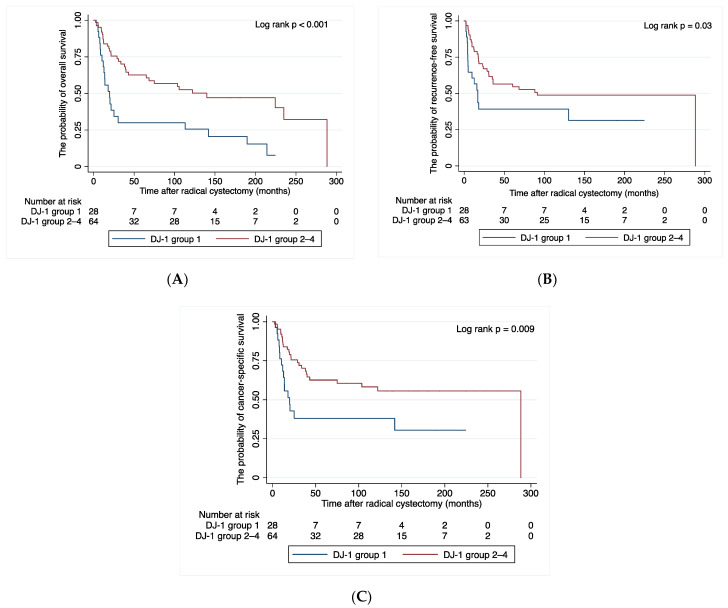
Probability of (**A**) overall survival, (**B**) recurrence-free survival, and (**C**) cancer-specific survival after radical cystectomy for Group 1 and Group 2–4. Group 1—cytoplasm-positive, nucleus-negative; Group 2—cytoplasm-positive, nucleus-positive; Group 3—cytoplasm-negative, nucleus-positive; Group 4—cytoplasm-negative, nucleus-negative.

**Table 1 cancers-14-02535-t001:** Relationship of DJ-1 serum concentration with the clinicopathologic characteristics of 172 patients.

Characteristic	Patients (*n*)	Serum DJ-1 (ng/µL)		*p* Value ^a^
		Median	Range	
Sex				
Men	135	21.5	0–235.1	
Women	37	27.2	0.81–207.0	0.37
Age group				
<65 Years	137	20.6	0–235.1	
≥65 Years	35	22.0	0–188.9	0.50
Pathology stage				
<pT1	129	23.5	2.3–235.1	
≥pT2	43	15.2	0–207.0	0.009
Pathology grade				
1, 2	122	22.4	1.8–235.1	
3	50	20.2	0–188.9	0.49
Lymph node status				
N0	166	22.0	0–235.1	
N1–2	6	16.1	7.6–29.9	0.34

^a^ By Mann–Whitney *U*-test, with 0.05 considered to be statistically significant.

**Table 2 cancers-14-02535-t002:** Association of DJ-1 expression with the clinicopathologic characteristics of patients who underwent radical cystectomy for bladder cancer.

Characteristic	Patients (*n*)	DJ-1 Expression (*n* (%))		*p* Value ^b^
		Group 1 ^a^	Groups 2–4 ^a^	
Total	92	28 (30.4)	64 (69.6)	
Sex				
Male	72 (78.3)	22 (30.6)	50 (69.4)	
Female	20 (21.7)	6 (30.0)	14 (70.0)	0.96
Age group				
<65	44 (47.8)	19 (43.2)	25 (56.8)	
≥65	48 (52.2)	9 (18.8)	39 (81.2)	0.011
Pathology stage				
pTa, pTis, pT1	20 (21.7)	2 (10.0)	18 (90.0)	
pT2–4	72 (78.3)	26 (36.1)	46 (63.9)	0.025
Pathology grade				
1, 2	36 (39.1)	10 (27.8)	26 (72.2)	
3	56 (60.9)	18 (32.1)	38 (67.9)	0.66
Carcinoma in situ				
Negative	80 (87.0)	24 (30.0)	56 (70.0)	
Positive	12 (13.0)	4 (33.3)	8 (66.6)	0.89
Lymph node status ^c^				
N0	65 (70.7)	17 (26.1)	48 (73.9)	
N1, N2	22 (23.9)	11 (50.0)	11 (50.0)	0.039
Lymphovascular invasion ^d^				
Negative	32 (37.6)	8 (25.0)	24 (75.0)	
Positive	53 (62.4)	19 (35.8)	34 (64.2)	0.30

^a^ Group 1—nucleus−, cytoplasm+; Group 2—nucleus+, cytoplasm+; Group 3—nucleus+ cytoplasm−; Group 4—nucleus−, cytoplasm−. ^b^ By chi-squared test, with 0.05 considered to be statistically significant. ^c^ Status was unknown in 5 patients. ^d^ Status was not available for 7 patients.

**Table 3 cancers-14-02535-t003:** Multivariate Cox proportional hazards regression analyses of DJ-1 expression and clinicopathologic findings for predicting clinical outcome after radical cystectomy.

Variable	Univariate Analysis			Multivariate Analysis		
	HR	95% CI	*p* Value	HR	95% CI	*p* Value
(A) Overall survival						
DJ-1 nucleus−, cytoplasm+	2.60	1.48 to 4.59	0.001	4.2	2.14 to 8.49	<0.001
Age (<65, ≥65)	1.11	0.64 to 1.92	0.72	1.83	0.94 to 3.55	0.075
Pathology stage	3.62	1.56 to 8.40	0.003	2.79	0.96 to 8.12	0.06
Pathology grade	1.84	1.01 to 3.35	0.046	1.29	0.66 to 2.55	0.46
Lymphovascular invasion	2.27	1.19 to 4.32	0.012	1.65	0.79 to 3.45	0.18
Lymph node metastasis	2.95	1.59 to 5.48	0.001	2.55	1.28 to 5.08	0.008
(B) Recurrence-free survival						
DJ-1 nucleus−, cytoplasm+	1.94	1.06 to 3.58	0.03	2.43	1.20 to 4.95	0.01
Age (<65, ≥65)	0.85	0.48 to 1.51	0.58	1.05	0.53 to 2.08	0.89
Pathology stage	2.64	1.12 to 6.25	0.027	2.03	0.67 to 6.14	0.21
Pathology grade	1.28	0.70 to 2.35	0.42	0.94	0.47 to 1.88	0.86
Lymphovascular invasion	2.66	1.31 to 5.43	0.007	1.87	0.83 to 4.22	0.13
Lymph node metastasis	3.21	1.73 to 5.98	< 0.001	2.60	1.31 to 5.16	0.006
(C) Cancer-specific survival						
DJ-1 nucleus−, cytoplasm+	2.28	1.21 to 4.30	0.01	3.53	1.61 to 7.57	0.001
Age (<65, ≥65)	0.90	0.49 to 1.67	0.75	1.25	0.60 to 2.63	0.55
Pathology stage	2.37	0.99 to 5.66	0.052	1.55	0.50 to 4.84	0.45
Pathology grade	1.39	0.73 to 2.65	0.32	1.02	0.48 to 2.14	0.96
Lymphovascular invasion	2.50	1.18 to 5.32	0.017	1.68	0.70 to 4.07	0.25
Lymph node metastasis	3.70	1.92 to 7.13	< 0.001	3.45	1.63 to 7.29	0.001

HR—hazard ratio; CI—confidence interval.

## Data Availability

The datasets used and/or analyzed during the present study are available from the corresponding author on reasonable request.

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
