# Peer review of "DJ-1 Expression Might Serve as a Biologic Marker in Patients with Bladder Cancer"

_cancers, 2022, doi:10.3390/cancers14102535_

Round 1

Reviewer 1 Report

Reviewer’s report for Cancers-1703029

There are no reservations for the publication of the manuscript entitled "DJ-1 Expression Might Serve as a Biologic Marker in Patients with Bladder Cancer" by Hirano et al in Cancers in its current revised form.

Author Response

Response to Reviewer 1 Comments

There are no reservations for the publication of the manuscript entitled "DJ-1 Expression Might Serve as a Biologic Marker in Patients with Bladder Cancer" by Hirano et al in Cancers in its current revised form.

Answer: Thank you so much for your comment. The present study is the first report to evaluate serum DJ-1 levels in patients with bladder cancer and the localization of DJ-1 expression in bladder cancer tissues. We hope that the present study will be a special interest to the readers of Cancers.

Reviewer 2 Report

The article greatly improved, thanks also to the addition of new Figures that are more appropriate than the previous ones.  The authors have answered most of my concerns. I would only suggest the following minor revisions:

1: I would recommend adding in the Methods the sensibility of the RPPA, applied to DJ-1 measures. Please, if not yet present, add also the references cited in the cover letter.

Finally, better refer in the discussion just to ELISA method, which permits more accurate and quicker measures than Western blots, and is most suitable for diagnostic purposes.

2: Please add part of the response and the cited reference in the Methods.

3-4. Even though samples are not the same, please discuss the finding that Serum DJ-1 expression levels but not the overall tissue levels are associated with OS and CCS. Maybe these results might be also discussed in relation to the finding that cytoplasmic enrichment of DJ-1 (group 1) in tumor cells is associated with OS and CCS. Which could be the mechanism by which DJ-1 is released in serum?

Author Response

Response to Reviewer 2 Comments

#1: I would recommend adding in the Methods the sensibility of the RPPA, applied to DJ-1 measures. Please, if not yet present, add also the references cited in the cover letter.

Finally, better refer in the discussion just to ELISA method, which permits more accurate and quicker measures than Western blots, and is most suitable for diagnostic purposes.

Answer: Thank you so much for your fruitful comments. RPPA is an emerging proteomics technology capable of validating new biomarkers because of its overwhelmingly high throughput [1, 2]. Furthermore, RPPA requires significantly smaller amounts of clinical samples for quantification than established clinical tests, such as ELISA [3]. Unfortunately, as you say, there is no report that the sensibility of the RPPA was demonstrated. We added the sentences in Materials and Methods section below. In term of ELISA, although this study did not investigate the differences of quantification between ELISA and Western blots, we agree with your comment. Again, thank you for your valuable comments.

Page4, “Materials and Methods”, Line 142-144.

Reverse-phase protein array require significantly smaller amounts of clinical samples for quantification than established clinical tests, such as ELISA [23].

#2:  Please add part of the response and the cited reference in the Methods.

Answer: Thank you so much for your valuable comments. We added the references in Materials and Methods section, and the references [4, 5].

Page4, “Materials and Methods”, Line 144-145.

We have been conducting and reporting on reverse-phase protein array analysis using micro-dot blot [22,24]

#3-4:  Even though samples are not the same, please discuss the finding that Serum DJ-1 expression levels but not the overall tissue levels are associated with OS and CCS. Maybe these results might be also discussed in relation to the finding that cytoplasmic enrichment of DJ-1 (group 1) in tumor cells is associated with OS and CCS. Which could be the mechanism by which DJ-1 is released in serum?

Answer: Thank you so much for your valuable comments. He et al reported serum DJ-1 in pancreatic cancer was not correlated with cancer progression and suggested that DJ-1 was secreted not only by cancer cells but also by cancer associated stroma cells [6]. Kawate et al reported breast cancer cells that showed low expression of DJ-1 proteins were associated with DJ-1 protein secretion [7]. They proposed that breast cancer cells secrete the DJ-1 isoform from the pool of intracellular DJ-1 into the serum. As you pointed out, it is important to reveal the mechanism that serum DJ-1 is secreted by bladder cancer cell, but it remains to be clarified. Notably, we would like to focus on serum DJ-1 as a detection marker in this study. However, your comment is very valuable so that we will accumulate the patients and resolve a mechanism in another study. In the present study, we just described limitation in Discussion section below.   

Page11, “Discussion”, Line361-363.

Fifth, different patient cohorts were used for the serum and immunohistochemistry analyses. A better understanding of any associations of serum DJ-1 with related immunohistochemistry findings remains to be established.

References

[1] Grote, T.; Siwak, D.R.; Fritsche, H.A.; Joy, C.; Mills, G.B.; Simeone, D.; Whitcomb, D.C.; and Logsdon, C.D. Validation of reverse phase protein array for practical screening of potential biomarkers in serum and plasma: accurate detection of CA19–9 levels in pancreatic cancer. Proteomics 2008, 8, 3051–3060. DOI: 10.1002/pmic.200700951

[2] Wulfkuhle, J.D.; Liotta, L.A.; Petricoin, E.F. Proteomic applications for the early detection of cancer. Nat. Rev. Cancer 2003, 3, 267–275.

[3] Matsubara, J.; Ono, M.; Honda, K.; Negishi, A.; Ueno, H.: Okusaka, T.; Furuse, J.; Furuta, K.; Sugiyama, E.; Saito, Y.; Kaniwa, N.; Sawada, J.; Shoji, A.; Sakuma, T.; Chiba, T.; Saijo, N.; Hirohashi, S.; Yamada, T.; Survival prediction for pancreatic cancer patients receiving gemcitabine treatment. Mol Cell Proteomics 2010, 9, 695–704. DOI: 10.1074/mcp.M900234-MCP200

[4] Kobayashi, M.; Nagashio, R.; Jiang, S.X.; Saito, K.; Tsuchiya, B.; Ryuge, S.; Katono, K.; Nakashima, H.; Fukuda, E.; Goshima, N.; et al. Calnexin Is a Novel Sero-Diagnostic Marker for Lung Cancer. Lung Cancer 2015, 90, 342-345. DOI: 10.1016/j.lungcan.2015.08.015

[5] Yanagita, K.; Nagashio, R.; Jiang, S.X.; Kuchitsu, Y.; Hachimura, K.; Ichinoe, M.; Igawa, S.; Nakashima, H.; Fukuda, E.; Goshima, N.; et al. Cytoskeleton-Associated Protein 4 Is a Novel Serodiagnostic Marker for Lung Cancer. Am J Pathol 2018, 188, 1328-1333. DOI: 10.1016/j.ajpath.2018.03.007

[6] He, X.Y.; Liu, B.Y.; Yao, W.Y.; Zhao, X.J.; Zheng, Z.; Li, J.F.; Yu, B.Q.; Yuan, Y.Z. Serum DJ-1 as a diagnostic marker and prognostic factor for pancreatic cancer. J Dig Dis 2011, 12, 131-137. DOI: 10.1111/j.1751-2980.2011.00488.x

[7] Kawate, T.; Iwaya, K.; Koshikawa, K.; Moriya, T.; Yamasaki, T.; Hasegawa, S.; Kaise, H.; Fujita, T.; Matsuo, H.; Nakamura, T.; et al. High levels of DJ-1 protein and isoelectric point 6.3 isoform in sera of breast cancer patients. Cancer Sci 2015, 106, 938-943. DOI: 10.1111/cas.12673

Reviewer 3 Report

Prognostic stratification according to DJ-1 staining patterns is hard to accept as a solid finding. In addition, serum DJ-1 levels do not correlate with tumour stage or tumour burden. This study is short in analysis on the molecular mechanism why Cy+/N- for DJ-1 expression is associated with aggressive clinical behaviour of bladder cancer. Although possible molecular mechanisms underlying clinical findings are discussed based on speculation from the literature, the authors need to show the possibility based on their own analyses.

Author Response

Response to Reviewer 3 Comments

Prognostic stratification according to DJ-1 staining patterns is hard to accept as a solid finding. In addition, serum DJ-1 levels do not correlate with tumour stage or tumour burden. This study is short in analysis on the molecular mechanism why Cy+/N- for DJ-1 expression is associated with aggressive clinical behavior of bladder cancer. Although possible molecular mechanisms underlying clinical findings are discussed based on speculation from the literature, the authors need to show the possibility based on their own analyses.

Answer: Thank you for your sharp opinion. We investigated several patterns for checking prognostic values in our study. According to these pilot studies, the present stratification showed the best value using statistical analyses in this study. In addition, we previously reported four expression patterns of S100A16, which is identical pattens of this study, in lung adenocarcinoma [1]. We believe that this stratification is acceptable for clinical analyses.

In terms of serum expression of DJ-1, He et al reported serum DJ-1 in pancreatic cancer was not correlated with cancer progression and suggested that DJ-1 was secreted not only by cancer cells but also by cancer associated stroma cells [2]. Kawate et al reported breast cancer cells that showed low expression of DJ-1 proteins were associated with DJ-1 protein secretion [3]. They proposed that breast cancer cells secrete the DJ-1 isoform from the pool of intracellular DJ-1 into the serum. In addition, Han et al. reported that serum DJ-1 were higher in patients with early stage of lung cancer than in those with advanced stage, consisted with the present study [4]. As you mentioned, it is necessary to suggest the possibility based our own analyses. Our unpublished data showed that DJ-1 knockdowned with siRNA substantially suppressed the invasive activity of T24 cells (human bladder cancer cell line) which showed DJ-1 overexpression by Western blots. As you pointed out, it is important to reveal the mechanism that serum DJ-1 is secreted by bladder cancer cell, but it remains to be clarified. Although unpublished in vitro experiment demonstrated a role of DJ-1, we would like to focus on serum DJ-1 as a detection marker and on expression of DJ-1 in pathological specimen as a prognostic marker in this study. However, your comment is very valuable so that we will accumulate the patients and resolve a mechanism more in another future study. In the present study, we just described limitation in Discussion section below. Again, thank you for your valuable comment and understanding the concept of the present study.

Page11, “Discussion”, Line361-363.

Fifth, different patient cohorts were used for the serum and immunohistochemistry analyses. A better understanding of any associations of serum DJ-1 with related immunohistochemistry findings remains to be established.

References

[1] Kobayashi, M.; Nagashio, R.; Saito, K.; Aguilar-Bonavides, C.; Ryuge, S.; Katono, K.; Igawa, S.; Tsuchiya, B.; Jiang, S.X.; Ichinoe, M.; et al. Prognostic significance of S100A16 subcellular localization in lung adenocarcinoma. Hum Pathol 2018, 74, 148-155. DOI:  10.1016/j.humpath.2018.01.001

[2] He, X.Y.; Liu, B.Y.; Yao, W.Y.; Zhao, X.J.; Zheng, Z.; Li, J.F.; Yu, B.Q.; Yuan, Y.Z. Serum DJ-1 as a diagnostic marker and prognostic factor for pancreatic cancer. J Dig Dis 2011, 12, 131-137. DOI: 10.1111/j.1751-2980.2011.00488.x

[3] Kawate, T.; Iwaya, K.; Koshikawa, K.; Moriya, T.; Yamasaki, T.; Hasegawa, S.; Kaise, H.; Fujita, T.; Matsuo, H.; Nakamura, T.; et al. High levels of DJ-1 protein and isoelectric point 6.3 isoform in sera of breast cancer patients. Cancer Sci 2015, 106, 938-943. DOI: 10.1111/cas.12673

[4] Han, B.; Wang, J.; Gao, J.; Feng, S.; Zhu, Y.; Li, X.; Xiao, T.; Qi, J.; Cui, W. DJ-1 as a potential biomarker for the early diagnosis in lung cancer patients. Tumour Biol 2017, 39, 1010428317714625. DOI: 10.1177/1010428317714625

Reviewer 4 Report

1. The authors do not respond to the question "In group 4, DJ-1 is all negative in the nucleus and cytoplasm. Group 4 may not be suitable for combining with groups 2 and 3."

Author Response

Response to Reviewer 4 Comments                                                                 

#1. The authors do not respond to the question "In group 4, DJ-1 is all negative in the nucleus and cytoplasm. Group 4 may not be suitable for combining with groups 2 and 3."

Answer: Thank you for your comment. Group 4 contained DJ-1 was all negative in the nucleus and cytoplasm of tumor cells. We do not think Group 4 may not be suitable for combining with Group 2 and 3. First, non-neoplastic tissues or peritumoral vascular endothelial cells are internal positive control so that DJ-1 without staining in the nucleus and cytoplasm does not show error of staining. Second, we previously reported four expression patterns of S100A16 in lung adenocarcinoma, including staining with all negative in the nucleus and cytoplasm [1]. We think DJ-1 without staining in the nucleus and cytoplasm is the true results as long as specimens contained internal positive control. According to them, we categorized DJ-1 without staining in the nucleus and cytoplasm into Group 4. I apologized for skipping your valuable comments.

References

[1] Kobayashi, M.; Nagashio, R.; Saito, K.; Aguilar-Bonavides, C.; Ryuge, S.; Katono, K.; Igawa, S.; Tsuchiya, B.; Jiang, S.X.; Ichinoe, M.; et al. Prognostic significance of S100A16 subcellular localization in lung adenocarcinoma. Hum Pathol 2018, 74, 148-155. DOI:  10.1016/j.humpath.2018.01.001

Round 2

Reviewer 3 Report

This study is still preliminary in that the authors did not addresss the molecular mechanism why Cy+/N- for DJ-1 expression is associated with aggressive clinical behaviour of bladder cancer. Although possible molecular mechanisms underlying clinical findings are discussed based on speculation from the literature, the authors need to show the possibility based on their own analyses.

This manuscript is a resubmission of an earlier submission. The following is a list of the peer review reports and author responses from that submission.

Round 1

Reviewer 1 Report

Reviewer’s report for Cancers-1617050

In the article entitled "DJ-1 Expression Might Serve as a Biologic Marker in Patients with Bladder Cancer", Hirano et al have studied serum expression levels and subcellular localization of DJ-1 in a cohort of 172 bladder cancer patients versus 20 patients with urolithiasis and 100 healthy individuals. They found that:

  • serum DJ-1 levels were higher in patients with BC than in those with urolithiasis or in healthy participants;
  • serum DJ-1 concentrations were significantly higher in patients with pTa/1 cancer than in those with muscle-invasive cancer;
  • DJ-1 overexpression in the cytoplasm and loss of nuclear expression was associated with poor prognosis in patients with BC.

After the year 2010, a few studies have focused on the expression levels of DJ-1 in bladder cancer, using urine samples, in order to establish a possible role of this putative cancer biomarker in diagnosis and prognosis. On the contrary, this work has focused on serum expression and subcellular localization of DJ-1, along with the usual studies associated to the clinicopathological characteristics of bladder cancer patients versus controls.

This is a well-written, compact and straightforward paper of interest. There are no reservations for the scientific approaches or the tools used and procedures followed in this work, so it may be published in Cancers. The only thing probably missing here, although mentioned in the Discussion, but only for urinary DJ-1 (It is conceivable that urinary DJ-1 on its own is insufficient as a diagnostic marker for BC) is a clear stress on the fact that serum DJ-1 validity as a bladder cancer biomarker is also limited if this is not applied along with additional relative biomarkers. This is apparent e.g. in Fig. 1, where the lower standard deviation values in cancer patients and healthy controls are more or less the same.

Reviewer 2 Report

The article aims to investigate serum DJ-1 level and DJ-1 expression and localization in tumor samples as diagnostic and prognostic BC biomarkers, respectively. Even though the article has the value to investigate for the first time DJ-1 serum expression levels and tumor intracellular localization in BC patients, the amount of data reported is a bit poor and there are key aspects that need further attention.

Major points:

  • Why do authors choose the reverse-phase protein array analysis instead of ELISA method used in other articles measuring DJ-1 serum or urine levels? Please comment about the choice, the sensitivity, and if the values measured in the healthy control groups are comparable among the different methods.
  • IHC on non-tumoral or peritumoral samples as control should be shown. Moreover, even though automatic immunostaining has been performed and two investigators have analysed the samples independently, I think that subdivision in groups 1 and 2 is quite difficult, whereas groups 3 and 4 are more easily recognizable. Since this is fundamental for all the following analysis and the main findings of the paper (a) I would recommend using a computational approach to calculate the sample score and assign the samples to a specific group; (b) I would ask authors to provide a panel with 4-5 examples for group 1, group 2, and some controversial cases, as well as the total number of the controversial cases and their final group allocation; (c) Please repeat the analysis also removing the initial controversial cases.
  • In the work by He and coauthors, the expression of DJ-1 was negatively correlated with lung cancer metastasis. This conclusion was supported at both the blood and tissue levels. In the work of Lee and coauthors (2012), on urothelial carcinoma, DJ-1 tissue overexpression was associated with invasive urothelial carcinoma. It would be interesting
  • to consider if in this study the expression level of DJ-1 in tumor tissues alone (independently of its localization in the nucleus and/or cytoplasm) correlates with clinicopathological features and/or survival.
  • to compare DJ-1 serum level directly with DJ-1 expression in tumor tissues. Is it possible to have both data in a subgroup of patients?
  • In any case, to discuss the relationship between the expression data on serum and tumor tissues.
  • Please add an ethical statement
  • The Discussion is fragmentary, with the same concept repeated several times. The first paragraph of the Discussion (lines 1-9), for example, might be canceled, since anticipates concepts that are better described later.
  • Recently, cancer research is focusing on the significance of different subcellular localization of oncogenic proteins as well as lncRNAs in tumor prognosis. Please discuss deeper this general aspect

Minor points:

  • In Figure 1B DJ-1 serum levels are higher in pT1BC than in pT2BC. In this respect, it would be interesting also to compare pT2BC directly to healthy control and urolithiasis groups.
  • Please briefly describe urotheliasis
  • The data reported in the last lines of paragraph 3.1 of Results (pag.5) are discordant with respect to Figure 2. Please correct.
  • Please provide details on the pilot analysis of overall survival in group 1, mentioned on page 6
  • In Table 2 p value for age is also significant (0.011), please comment.
  • The summary needs some editing
  • On page 2, at the end of the introduction authors report: “Few studies have demonstrated that DJ-1 acts as a serum marker for the detection of bladder cancer”. Please add references

Reviewer 3 Report

The authors investigated a diagnostic role of serum DJ-1 levels and prognostic roles of its expression in tumor tissues among bladder cancer patients. The authors demonstrated that serum DJ-1 was higher in bladder cancer patients than in non-UC controls and that positive cytoplasmic but negative nuclear DJ-1 expression was associated with adverse prognosis than other expression patterns in bladder cancer patients undergoing radical cystectomy. The authors concluded that DJ-1 expression may help physicians decide on further follow-up and additional treatments.   This paper is too descriptive and does not address mechanistic basis underlying clinical findings obtained. What do the authors consider on the role of cytoplasmic DJ-1 expression? What about that of nuclear DJ-1 expression? The authors need to discuss them from the aspects of both their IHC data and biological functions of DJ-1.   The following comments may help improve the present manuscript. 1_ Table 1: Despite description in the text, no information on healthy and urolithiasis controls was provided. 2_ The authors may want to address the biological roles of cytoplasmic versus nuclear expression of DJ-1 in bladder cancer. The authors may want to explain why Cy+/Nuc- expression of DJ-1 is associated with worst prognosis. To address this, the authors may want to evaluate prognostic effects of Cy+ and Nuc+ separately using H-score as a continuous variable. 3_ Multivariable prognosis analysis: Age should be included in the analyses because of its significant association with DJ-1 expression in bladder cancer tissues. Too many variables were included in MVA for the number of events according to the general rule of one MVA variable for 10 events, which may result in overfitting of multivariable models. 4_ The reviewer wonders if serum DJ-1 can really play a role as diagnostic biomarker of bladder cancer because serum DJ-1 does not seem to be associated simply with tumor stage. What about associations of serum DJ-1 and histological grade and tumor volume? 

Reviewer 4 Report

  1. How are sera stored in the hospital?
  2. The authors do not provide the institutional approval number.
  3. In the 2.1 patients section, “The 172 patients included in the study were those who remained after cancers other than UC” is hard to understand.
  4. In Table 1 and Fig. 1B, P-value is not consistent.
  5. In AUC analysis, line 2, “urolithiasis area under the curve [AUC], 0.88”, AUC should be 0.83.
  1. In detecting low-stage bladder cancer, ROC analysis can be performed with pT1 and healthy persons.
  2. In group 4, DJ-1 is all negative in the nucleus and cytoplasm. Group 4 may not be suitable for combining with groups 2 and 3.
  3. In Table 2, only age and stage are associated with DJ-1 expression. However, in univariate analysis, stage, grade, invasion, and metastasis are correlated with survival, RFS, and CSF. The authors may explain these phenomena.